# Distinct Urinary Proteome Changes Across Estimated Glomerular Filtration Rate Stages in a Cohort of Black South Africans

**DOI:** 10.3390/ijms26041740

**Published:** 2025-02-18

**Authors:** Siyabonga Khoza, Jaya A. George, Previn Naicker, Stoyan H. Stoychev, Rethabile J. Mokoena, Ireshyn S. Govender, June Fabian

**Affiliations:** 1Department of Chemical Pathology, National Health Laboratory Service, Faculty of Health Sciences, University of the Witwatersrand, Johannesburg 2000, South Africa; 2National Health Laboratory Service, Johannesburg 2192, South Africa; 3Academic Affairs, Research & Quality Assurance, National Health Laboratory Service, Johannesburg 2000, South Africa; 4Wits Diagnostic Innovation Hub, University of the Witwatersrand, Johannesburg 2000, South Africa; 5ReSyn Biosciences, Edenvale 1610, South Africa; 6Evosep Biosystems, 5230 Odense, Denmark; 7Future Production Chemicals, Council for Scientific and Industrial Research, Pretoria 0001, South Africa; 8Wits Donald Gordon Medical Centre, School of Clinical Medicine, Faculty of Health Sciences, University of the Witwatersrand, Johannesburg 2000, South Africa; 9South African Medical Research Council/Wits University Rural Public Health and Health Transitions Research Unit (Agincourt), School of Public Health, Faculty of Health Sciences, University of the Witwatersrand, Johannesburg 2000, South Africa

**Keywords:** chronic kidney disease, estimated glomerular filtration, albuminuria, urinary proteomics, biomarker

## Abstract

Kidney function parameters including estimated glomerular filtration rate (eGFR) and urine albumin excretion are commonly used to diagnose chronic kidney disease (CKD). However, these parameters are relatively insensitive, limiting their utility for screening and early detection of kidney disease. Studies have suggested that urinary proteomic profiles differ by eGFR stage, offering potential insights into kidney disease pathogenesis alongside opportunities to increase the sensitivity of current testing strategies. In this study, we characterized and compared the urinary proteome across different eGFR stages in a Black African cohort from rural Mpumalanga Province, South Africa. We stratified 81 urine samples by eGFR stage (mL/min/1.73 m^2^): Stage G1 (eGFR ≥ 90; *n* = 36), Stage G2 (eGFR 60–89; *n* = 35), and Stage G3–G5 (eGFR < 60; *n* = 10). Urine proteomic analysis was performed using an Evosep One liquid chromatography system coupled to a Sciex 5600 TripleTOF in data-independent acquisition mode. Nonparametric multivariate analysis and receiver operating characteristic (ROC) curves were used to assess the performance of differentially abundant proteins (DAPs). Pathway analysis was performed on DAPs. Creatinine-based eGFR was calculated using the Chronic Kidney Disease Epidemiology Collaboration (CKD-EPI) equation. In this study, thirty-eight urinary proteins were differentially abundant for eGFR Stages 3–5 when compared to Stages G1 (AUC = 0.95; CI: 0.86–1) and G2 (AUC = 0.84; CI: 0.64–0.98). Notably, only six urinary proteins (Cystatin M (CST6), glutathione hydrolase 6 (GGT6), sushi domain containing 2 (SUSD2), insulin-like growth factor binding protein 6 (IGFBP6), heat shock protein 90 beta family member 1 (HSP90B1), and mannosidase alpha class 1A member 1 (MAN1A1)) were differentially abundant when comparing Stage G1 and Stage G2 with a modest AUC = 0.81 (CI: 0.67–0.92). Pathway analysis indicated that DAPs were associated with haemostasis and fibrin clot formation. In a rural cohort from South Africa, the urinary proteome differed by eGFR stage, and we identified six differentially abundant proteins which, in combination, could help to differentiate earlier eGFR stages with higher predictive accuracy than the currently available tests.

## 1. Introduction

Chronic kidney disease (CKD) is defined as the presence of kidney damage or decreased kidney function for at least three months, which is detrimental to health [1]. Kidney damage can be established by biopsy, imaging, or laboratory markers such as the presence of haematuria, abnormalities in urine sediments, and albuminuria (urine albumin: creatinine ratio (uACR) ≥ 3 mg/mmol). Decreased kidney function refers to a reduced glomerular filtration rate (GFR), which is frequently estimated from serum creatinine [1]. As per the 2024 Kidney Disease Improvement Global Outcomes (KDIGO), using eGFR (mL/min/1.73 m^2^), CKD is classified into five stages (G1–G5) which are G1 (≥90), G2 (60–89), G3a (45–59), G3b (30–44), G4 (15–29), and G5 (<15) [1]. Using albuminuria, measured as the urine albumin excretion rate (uACR measured as mg/mmol), CKD is categorized into three groups: A1 (<3), A2 (3–30), and A3 (≥30) [1]. Despite their high utilization as markers to diagnose and monitor CKD, both uACR and eGFR have their limitations. Unfortunately, creatinine-based eGFR is confounded by factors such as muscle mass, age, and sex, with high intra-individual and inter-individual variation in serum creatinine [2]. Additionally, creatinine-based eGFR has poor predictive ability for the risk of CKD progression and adverse all-cause and cardiovascular outcomes [3,4].

When looking at alternative biomarkers, the urinary proteome seems to differ by eGFR stage in terms of the differential abundance and number of proteins detected [5,6]. In a study comparing Stage G1–3 CKD patients to healthy controls in an Asian population, more urinary proteins were expressed among the healthy controls compared to those with CKD, with lower urinary protein expression occurring in those with advanced stages of CKD, and urinary proteins such as beta-2 microglobulin, Fetuin A, vitamin-D binding globulin, and alpha-1 microglobulin/bikunin precursor were linked to stages with poorer kidney function [5]. A comparison of early-stage and late-stage CKD revealed 929 significantly differentially expressed peptides, the majority of which were of collagen origin and downregulated [6]. When these peptides were combined as a CKD classifier, the scores from this classifier correlated well with eGFR stages [7]. When comparing different CKD stages (CKD stages 1, 2, and 5) to healthy controls, urine proteomics profiling differentiated early CKD stages from both advanced CKD stages and the control group. Additionally, the number of proteins and their patterns varied with increasing CKD severity [8]. At the protein level, cathepsin D, metalloproteinase 7, and insulin-like growth factor binding proteins were associated with decreased eGFR [9]. In severe stages of CKD (<29 mL/min/1.73 m^2^), blood-derived peptides such as alpha-1 antitrypsin, vitamin-D binding proteins, transthyretin, and serum amyloid A-1 were found to be highly abundant in the urinary proteome, while in early CKD stages, analysis showed modification of collagen fragments, which suggests changes in the turnover of the extracellular matrix (ECM), the principal structure of fibrotic tissue [10]. Furthermore, increased abundance was observed in fibronectin, a component of the ECM, as GFR declined [11]—suggesting a correlation between certain proteins and eGFR stages. However, most of the existing studies either compare advanced CKD with healthy controls or are limited by small sample sizes (with samples of five and nine patients) of those with early CKD [8,12,13]. Urinary proteomic profiling can help identify biomarkers of kidney dysfunction at the earliest stage of CKD, even before traditional markers such eGFR and albuminuria show noticeable changes. This enables timely intervention and better management of the disease [14]. Therefore, in this study, we aimed to generate and compare urinary proteome profiles stratified by eGFR stages in a Black South African population.

## 2. Results

### 2.1. Demographic and Clinical Characteristics

The demographic and clinical characteristics of the study participants are summarised in Table 1. Median age, sex, and BMI distributions were not statistically different between the groups. As expected, there was a progressive increase in uACR (*p* < 0.001) measurements with a decline in kidney function across the eGFR stages. Participants with Stage G3–G5 were more likely to have hypertension (*p* = 0.023) and diabetes (*p* = 0.004).

### 2.2. Performance of Study-Specific Suitability–Quality Control

Appendix A displays the performance of study-specific process control (CR) with a pooled urine sample from multiple patients, which was used to monitor the consistency of the sample preparation. As well as the system suitability controls (SSC), commercially obtained and pre-digested HeLa lysate was used to monitor the stability of the liquid chromatograph mass spectrometry (LCMS) system. The coefficient of variation (CV) at the protein group level was 15% for SSC and 14% for CR over 5 days of analysis. The CV at the peptide level was 18.2% and 17.5% for SSC and CR, respectively. Counts for protein, peptide, and precursors remained consistent throughout the data acquisition process (Appendix A) and within the acceptable level of ≤20% that is currently recommended for a proteomics workflow [15].

### 2.3. Multivariate Analysis of Differential Abundant Proteins (DAPs)

A total of 1469 proteins were identified in this cohort, and 38 urine proteins were found to be significantly different across the three stages using the Kruskal–Wallis H test (Figure 1 and Appendix A). Pairwise comparisons using Dunn’s test with a Bonferroni adjustment indicated that Stage G3–G5 scores were significantly different from those of Stage G1 and Stage G2 for all 38 differentially abundant proteins (Appendix A). Six urine proteins (CST6, GGT6, SUSD2, IGFBP6, HSP90B1, and MAN1A1) showed statistically significant differences between Stage G1 and Stage G2 (Figure 2 and Appendix A). To assess the discriminatory power of the selected proteins across all three stages, a multivariate approach was used. An unsupervised analysis using PCA did not show a clear separation between the stages (Figure 3 and Appendix A). The score plots overlapped, with all groups showing great dispersion, but Stage G3–G5 showed minimal dispersion.

Spearman’s correlations between serum creatinine concentrations, eGFR, and uACR with the 38 differentially abundant proteins are shown in Appendix A. Serum creatinine concentrations had a moderate positive correlation with CST6 (r = 0.43, *p* < 0.001) and correlated negatively with HSP90B1 (r = −0.37; *p* < 0.001), MAN1A1 (r = −0.30, *p* < 0.001), SUSD2 (r = −0.44, *p* < 0.001), and GGT (r = −0.39, *p* < 0.001) in all stages. There was a weak negative correlation between IGFBP6 and serum creatinine, and this correlation was not statistically significant (r = −0.026, *p* = 0.819). Additionally, negative correlations were observed between uACR and HSP90B1 (r = −0.31; *p* = 0.003), as well as MAN1A1 (r = −0.27, *p* = 0.016). No significant correlation was found between uACR and CST6 (r = 0.11, *p* = 0.332), SUSD2 (r = −0.18, *p* = 0.096), GGT (r = −0.18, *p* = 0.101), or IGFBP6 (r = 0.17, *p* = 0.141). Significant correlations were observed between eGFR and MAN1A1 (r = 0.37, *p* < 0.001), SUSD2 (r = 0.53, *p* < 0.001), CST6 (r = −0.33, *p* = 0.003), GGT (r = 0.46, *p* < 0.001), and HSP90B1 (r = 0.43, *p* < 0.001). No significant correlation was observed between IGFBP6 and eGFR (r = 0.05, *p* = 0.663). Overall, HSP90B1 was most consistently correlated with both eGFR (positively) and uACR (negatively).

To evaluate the diagnostic potential of the 38 differentially abundant proteins identified in this study, ROC curves were generated based on a linear support vector machine logistic algorithm including the predicted class probabilities with cross-validation. One hundred cross-validations were performed (Figure 4 and Figure 5) and their results were averaged to generate the plot. Each sample was predicted from the one hundred cross-validations. The ROC curve was further generated to evaluate group intercomparison. The area under curve (AUC) values ranged from 0.81 to 0.95, confirming a good fit of the model. The highest AUC (0.95, CI: 0.87–1) was achieved when comparing Stage G1 with Stage G3–G5. The predicted class probabilities for each sample are shown in Figure 4 and Figure 5. There was no absolute separation, but most of the samples could be distinguished correctly, especially between Stage G1 and Stage G3–G5, suggesting the misclassification of some samples in this cohort. The top urine proteins contributing to the prediction model were ranked by mean of importance (Figure 4B,F and Figure 5B). The discriminating ability between Stage G1 and Stage G2 was modest (AUC = 0.81; 95% CI: 0.67–0.92).

### 2.4. Pathway and Network Analysis of Differentially Abundant Proteins

The functional enrichment analysis and gene ontology of DAPs are shown in Figure 6 and Appendix A. To identify the functional pathways linked to our proteomic signatures and potentially associated with CKD progression, we conducted pathway enrichment analysis through the Enrichr database. The top five significantly (using adjusted *p* value < 0.05) enriched terms were the following: haemostasis (SERPINA5, SOD1, IGLL1, FAM3C, EGF, and SERPINC1) as the highest mechanism in the pathogenesis of CKD, the intrinsic pathway of fibrin clot formation (SERPINA5 and SERPINC1), the common pathway of fibrin clot formation (SERPINA5 and SERPINC1), keratin sulphate biosynthesis (B4GAT1 and OMD), and regulation of insulin growth factor 1 (IGF-1) transport and uptake by insulin growth factor binding proteins (HSP90B1, IGFBP6, and SERPINC1). Furthermore, IGHG3, B2M, and SOD1 were linked to the retina homeostasis (GO:0001895) biological process (Table 2). The ESCRT III Complex Disassembly (GO:1904903) biological process was associated with IST1 and VTA1 proteins, while the molecular function of DAPs was predominantly linked to inhibition of the endopeptidases.

## 3. Discussion

Using a data-independent acquisition (DIA) approach, the present study profiled urinary proteins to provide clues not only about pathophysiological mechanisms, but also potential proteins related to kidney function according to eGFR stage. 

To identify possible markers that differ according to eGFR stage,, we first explored differentially abundant proteins (DAPs) between Stage G3–G5 and the earlier stages, G1 and G2. Our results showed that DAPs changed depending on eGFR stage. Thirty-eight proteins were differentially abundant between Stage G3–G5 and the other groups. Among these, proteins such as SERPINA5, beta-2 microglobulin, and osteomodulin (Figure 1) were found to be differentially abundant in this study and have been extensively studied and linked to adverse outcomes in CKD [12,16,17]. However, to detect changes in the early stages of CKD, we focused on DAPs that were significantly different between participants in Stage G1 and Stage G2. Six proteins were identified as such: CST6, IGFBP6, MAN1A1, SUSD2, HSP90B1, and GGT6.

For stage G3–G5,, CST6 levels were significantly higher than in the Stage G1 and Stage G2, and correlated negatively with eGFR, suggesting that as kidney function declines (indicated by increased creatinine), CST6 levels increase. This relationship could indicate that CST6 might be involved in kidney injury or dysfunction, potentially serving as a marker for kidney damage or impairment. CST6, like Cystatin-C, belongs to the type 2 cystatin superfamily and is an extracellular polypeptide inhibitor of the cysteine proteases that prevent extra proteolysis. Proteases such as Cathepsins B, S, and L, as well as Legumain, are involved in kidney matrix remodelling and are inhibited by CST6 in humans, which may result in dysregulation of either their expression or activity [18,19,20]. These proteases are involved in the regulation of extracellular matrix homeostasis, apoptosis, glomerular permeability, endothelial function, and inflammation. Dysregulation of these proteases has been associated with the onset and progression of CKD [21]. Despite the link between CST6 and protease functioning, the exact mechanism that affects kidney disease has not been described.

We observed higher IGFBP6 protein levels in the Stage G3–G5. This is supported by several studies that have shown that IGFBP6 levels are frequently upregulated in CKD. The abundance of IGFBP6 in urine gradually increases with declining kidney function [22]. The pathophysiological role of IGFBP6 in CKD has not yet been established. The association between IGFBP6 and CKD is believed to be involved in kidney fibrosis through the regulation of apoptosis in kidney cells [23].

In this study, MAN1A1 and SUSD2 gradually decreased with decreasing kidney function, and their levels became lower with the increasing severity of kidney non-function, suggesting that these may play a role in kidney function or repair. Their reduced levels in patients with high uACR could reflect deteriorating kidney function. SUSD2 is a type I membrane protein containing domains of adhesion molecules [24]. The highest levels of SUSD2 are found in normal lung tissue, followed by kidney tissue, but no known function has been described in the kidneys. Few studies have reported an association between SUSD2 and different malignancies such as lung cancer or breast and renal cell carcinoma [25,26], where it acts as a tumour suppressor gene, with low levels being associated with aggressiveness [27]. In the kidney, it is one of the urine proteins that are significantly decreased in acute kidney rejection after transplantation compared to controls [28], suggesting that low urine levels may be related to CKD. It should be noted that SUSD2 dysregulation is not entirely specific to kidney disease—abnormal levels have been identified in malignancies as well [25]. While no other studies have linked MAN1A1 to CKD, in knockout mice the MANA1B gene, which is closely related to MAN1A1, has shown an increased severity of acute and chronic kidney disease [29].

Data on the role of HSP90B1 are scarce, and its role in CKD is not fully understood. However, it is believed to be involved in cell survival and to act as a protein chaperone involved in protein folding and stabilisation [30]. Furthermore, it is involved in the maintenance of normal kidney blood flow and affects GFR by regulating the synthesis of nitric oxide dependent on endothelial NO-synthase [31]. In hypoxic conditions, it has important stress response functions in wound repair and healing [32]. HSP90B1 has also been found to mediate communication between B-7 and LRP5/β-catenin signalling in podocyte injury [33].

Glutathione hydrolase 6 (GGT6) plays an important role in cellular detoxification through the conjugation of glutathione to endogenous and exogenous compounds for elimination. It is also involved in the elimination of reactive oxygen species and xenobiotics. Therefore, dysregulation of glutathione activity is likely to affect the renal antioxidant defence system [34]. Since CKD is a state of oxidative stress [35], it may, therefore, be associated with a depletion of GGT6. Abnormalities in the expression of GGT6 are expected. Generally, GGT6 is low or undetectable in its normal state. However, low urinary GGT6 may suggest a depletion of glutathione levels, reflecting impaired antioxidant capacity and an intensified oxidative stress burden. It could also reflect the extent of fibrosis and structural damage in the kidneys [36,37].

Gene ontology analysis confirmed the pathways associated with CKD pathogenesis, such as haemostasis [38]. Renal insufficiency is associated with inflammation, which has a direct effect on haemostasis [39]. High hyperactivity of platelets has negative consequences on kidney function through coagulation cascade and/or fibrinolytic system activation [5,39,40,41], and these were among the highly enriched pathways in this study. Hyperactive platelets have been associated with stimulation of the inflammatory processes in CKD [42]. Coagulation abnormalities are the result of complex interactions between uraemic toxins, morphological changes in the walls of blood vessels, and platelet activity [39]. Coagulation abnormalities are associated with high thrombotic and haemorrhagic risk in kidney disease [43]. In addition to kidney disease being an inflammatory state, dysregulation of the coagulation system is associated with inflammatory processes [44,45]. Understanding the pathways related to CKD may enable early therapeutic intervention and mitigate its progression.

Our study has several limitations. Firstly, the study had a limited number of patients in the Stage G3–G5 category. In clinical practice, patients at advanced stages of CKD can easily be identified by traditional markers; therefore, the low number of patients in this category are unlikely to affect the findings of the study. Secondly, the study was conducted using a single cohort; therefore, the results may not be generalisable.

## 4. Materials and Methods

### 4.1. Ethics Statement

The study was approved (clearance number: M210128) by the Faculty of Health Sciences Human Research Ethics Committee (Medical) at the University of the Witwatersrand, Johannesburg, South Africa.

### 4.2. Sample Selection

Samples for this sub-study were selected from the South African arm of the African Research on Kidney (ARK) study, which aimed to determine the population prevalence of CKD and its associated risk factors in South Africa, Malawi, and Uganda. The detailed methods employed in the ARK study have previously been published [46]. The demographic data, clinical information, and laboratory results from the ARK study were accessed for inclusion in the current study, and data were de-identified prior to sharing. Urine samples collected during the ARK study were used for proteomics analysis. In this study, 81 participants were selected and categorized into three groups based on their estimated glomerular filtration rate (eGFR, mL/min/1.73 m^2^): Stage G1 (*n* = 36), Stage G2 (*n* = 35), and Stage G3–G5 (*n* = 10). Secondly, participants were age-matched (±5 years) and sex-matched between the groups where feasible. Additionally, we determined participants to be Stage G3–G5 if eGFR was confirmed as <60 mL/min/1.73m^2^ for ≥3 months. Those without confirmed low eGFR were not included. In the ARK study, participants were defined as being hypertensive (systolic blood pressure (SBP) ≥ 140 mm Hg and/or diastolic blood pressure (DBP) ≥ 90 mm Hg) based on the 7th Report of the Joint National Committee on Prevention, Detection, Evaluation, and Treatment of High Blood Pressure [47,48]. Diabetes was defined based on a non-fasting glucose measurement ≥ 11.1 mmol/L, and human immunodeficiency virus (HIV) status was defined as positive if a participant had previously been tested and knew their status or they were screened and tested during the study [48].

### 4.3. Clinical Laboratory Tests

Serum and urine creatinine was measured using Jaffe’s method, traceable to isotope dilution mass spectrometry, and urine albumin was measured using immunoturbidimetry through a Roche Cobas analyser (Roche Diagnostics, Mannheim, Germany). The estimated glomerular filtration rate (eGFR) was calculated from serum creatinine using the Chronic Kidney Disease Epidemiology Collaboration (CKD-EPI) equation from 2009, without adjustment for African American ethnicity [49].

### 4.4. Urine Protein Extraction

An in-house method was used for urinary proteome preparation as previously described [16,17]. Briefly, 1600 μL of ice-cold 80% acetone was added to 400 μL of urine. The samples were kept at −20 °C for 1hr, and then centrifuged at 12,000× *g* for 30 min. The supernatant was removed, and the pellets were dried using a 70 °C heating block (AccuBlock Digital dry bath, Labnet International, Inc., Edison, NJ, USA) for 1 min. The pellets were resuspended in 100 μL of 2% sodium dodecyl sulfate (SDS) and sonicated for 5 min. Proteins were reduced with 1 μL of 1M dithiothreitol (DTT) at 70 °C for 15 min and thereafter transferred to a 40 °C heating block for an additional 15 min, followed by alkylation with 6 μL of 500 mM iodoacetamide (IAA) for 30 min at room temperature (RT) in the dark. The proteins were digested on-bead using MagResyn^TM^ HILIC microparticles (ReSyn Biosciences, Edenvale, South Africa) using an automated KingFisher^TM^ Duo (Thermo Fisher Scientific, Rockford, IL, USA), as previously described [16,17]. The peptides were dried with a CentriVap vacuum concentrator (Labconco, Kansas City, MO, USA) overnight and resuspended in 40 μL of 2% acetonitrile/0.2% formic acid, then they were stored at −80 °C until LC–MS/MS analysis. Peptide quantification was performed using the Pierce™ Quantitative Colorimetric Peptide Assay (Thermo Fisher Scientific, Waltham, MA, USA). A pooled sample from 10 urine samples was prepared and analysed alongside individual samples as a study-specific process control, as explained above. Additionally, a commercial Hela digest system suitability control (SSC) was analysed.

### 4.5. Liquid Chromatography–Mass Spectrometric (LCMS) Analysis and Data Extraction

Digested peptides were analysed using an Evosep One LC system (Evosep ApS, Odense, Denmark) coupled to a SCIEX TripleTOF 5600 tandem mass spectrometer (Sciex, Framingham, MA, USA) in data-independent acquisition (DIA) mode. An Evosep performance column (EV1112, 15 cm × 75 μm, 1.9 μm) was used for the Whisper 40SPD method. The source settings of the Nanospray 3, which was equipped with a 20 µm Lotus emitter (Fossil Ion Technologies, Valencia, Spain), were as follows: CUR-20, GS1-30, ISVF-2900. Data were acquired using 48 MS/MS scans of overlapping sequential precursor isolation windows (variable *m*/*z* isolation width, 1 *m*/*z* overlap, high sensitivity mode), with a precursor MS scan for each cycle. The accumulation time was 50 ms for the MS1 scan (from 400 to 1100 *m*/*z*) and 30 ms for each product’s ion scan (200 to 1800 *m*/*z*) over a cycle time of 1.53 s.

A spectral library was built with Spectronaut^TM^ 19 software (Biognosys Schlieren, Schlieren, Switzerland) using the default settings with minor adjustments, including segmented regression for retention time (iRT), a Trypsin digestion rule, and acceptance of modified peptides with 3–6 intense fragments between 300 and 1800 *m*/*z*. This study-specific spectral library was combined with an in-house generated urinary proteome spectral library (using the Spectronaut™ “Search Archives” feature). Raw (. wiff) data files were analysed using Spectronaut™ 19 with the default settings for analysis. These default settings included the following: dynamic iRT retention time prediction with a correction factor for window 1; mass calibration was set to local; decoy method was set as scrambled; false discovery rate (FDR), according to the mProphet approach [50], was set at 1% on the precursor, peptide, and protein group levels; protein inference was set to “default”, which is based on the ID picker algorithm [51]; and global cross-run normalization on the median was selected. The final urinary proteome spectral library (peptides—20,616, protein groups—2604) was used as a reference for data extraction.

### 4.6. Statistical Analysis

The demographic and clinical characteristics were analysed using STATA 18SE (Stata Corp, College Station, TX, USA). All the data were not normally distributed; therefore, nonparametric tests were used, and a *p*-value < 0.05 was considered statistically significant. Categorical and continuous data were analysed using the chi-square test/Fisher’s exact test and the Kruskal–Wallis H test, respectively. Several nonparametric multivariate analyses were performed. Firstly, principal component analysis (PCA, unsupervised) was performed using the free web-based multivariate analysis tool Metaboanalyst v6.0 (https://www.metaboanalysts.ca/ (accessed on 20 May 2024)). To assess significant differences in the differentially abundant proteins (DAP) between the three groups, the Kruskal–Wallis H test, followed by post hoc analysis using Dunn’s test with a Bonferroni adjustment, was applied using GraphPad Prism 10 (GraphPad Software, San Diego, CA, USA). For this analysis, between-groups comparison was applied as follows: (1) Stage G1 vs. Stage G3–G5 (group 1); (2) Stage G2 vs. Stage G3–G5 (group 2); and Stage G1 vs. Stage G2 (group 3). Spearman’s correlation analysis between laboratory parameters and proteomic data was performed using STATA 18SE with Bonferroni adjustments. Receiver operating characteristic curves were constructed to predict the ability of selected proteins to classify patient groups using Metaboanalyst v6.0. Pathways and Gene Ontology (GO) were demonstrated using Enrichr/Enrichr-KG—https://maayanlab.cloud/Enrichr/—using the Reactome library 2022 (accessed on 20 May 2024). The top enriched pathways (*p* < 0.05) were selected.

## 5. Conclusions

In conclusion, we identified six urine proteins that were differentially abundant, between 60–89 and ≥90, depending on eGFR stage. These proteins, when combined into a six-protein model, showed that it could discriminate early CKD stages from late stages with high predictive accuracy. This study provides evidence that several unique proteins are involved in the early and late stages of CKD. It further suggests that a selected combination of biomarkers could be used to stratify patients into different CKD stages. Bioinformatics analysis highlighted the involvement of haemostasis and abnormalities in fibrin formation pathways in CKD’s pathophysiology, possibly influencing the disease’s progression.

## Figures and Tables

**Figure 1 ijms-26-01740-f001:**
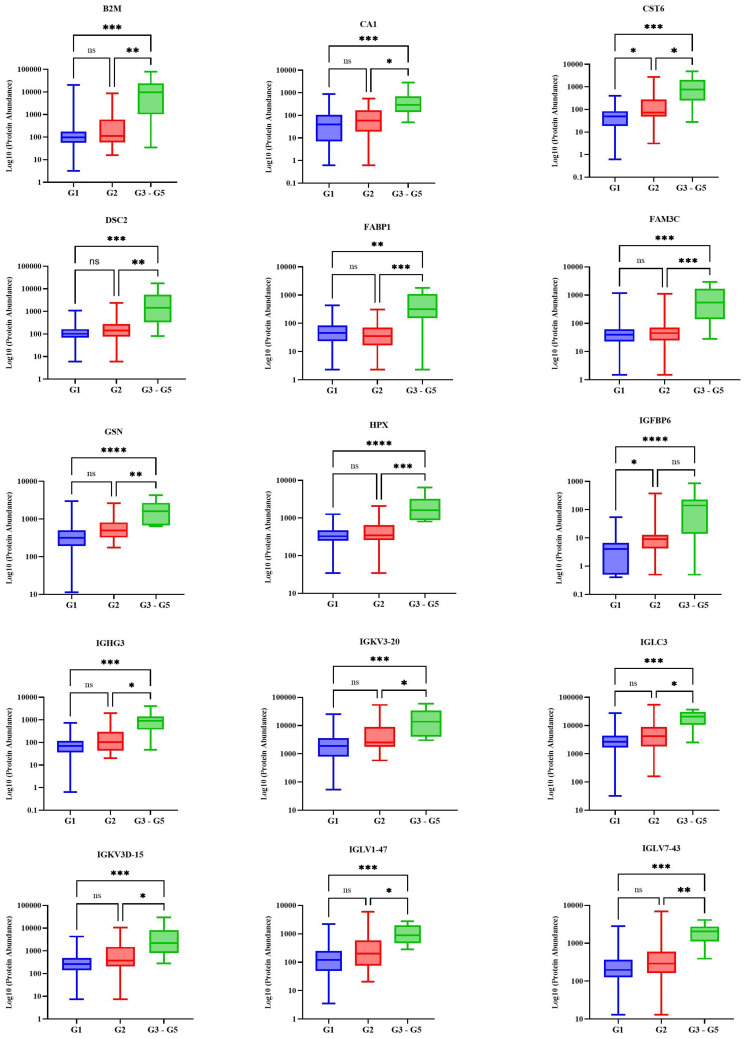
Box and whisker plots of the proteins that were significantly different between stages after a Kruskal–Wallis test followed by a Dunn’s test showing the urine proteins expressed in a logarithmic scale (log10) at Stage G1 (eGFR ≥ 90 mL/min/1.73 m^2^), Stage G2 (eGFR 60–89 mL/min/1.73 m^2^), and Stage G3–G5 (eGFR < 60 mL/min/1.73 m^2^). Beta-2 microglobulin (B2M), Carbonic anhydrase 1 (CA1), Cystatin M (CST6), Desmocollin-2 (DSC2), Fatty acid-binding protein, liver (FABP1), FAMC metabolism regulation signalling molecule C (FAM3C), Gelsolin (GSN), Hemopexin (HPX), Insulin-like growth factor binding protein 6 (IGFBP6), Immunoglobulin heavy constant gamma 3 (IGHG3), Immunoglobulin kappa variable 3-20 (IGK3-20), Immunoglobulin lambda constant 3 (IGLC3), Immunoglobulin kappa variable 3D-15 (IGKV3D-15), Immunoglobulin lambda variable 1-47 (IGLV1-47), Immunoglobulin lambda variable 7-43 (IGLV7-43), Immunoglobulin kappa light chain (P0DOX7), Peptidase inhibitor 16 (PI16), Antithrombin III (SERPINC1), Superoxide dismutase (SOD1), Selotransferrin (TRFE), Immunoglobulin lambda-like polypeptide 1 (IGLL1), Immunoglobulin lambda-1 light chain (P0DOX8), Beta-1,4-glucuronyltransferase 1 (B4GAT1), Neural cell adhesion molecule L1-like protein (CHL1), Pro-epidermal growth factor (EGF), Heat shock protein 90 beta family member 1 (HSP90B1), IST1 homolog (IST1), mannosidase alpha class 1A member 1 (MAN1A1), Multimerin-2 (MMRN2), Osteomodulin (OMD), Phosphatidylcholine-sterol acyltransferase (LCAT), Plasma serine protease inhibitor (SERPINA5), Sushi domain-containing protein 2 (SUSD2), Glutathione hydrolase 6 (GGT6), Vacuolar protein sorting-associated protein VTA1 homolog (VTA1), and Heat shock protein HSP 90-beta (HSP90A1). Asterisks indicate statistically significant differences between the stages, where **** *p* < 0.0001, *** *p* < 0.001, ** *p* < 0.01, and * *p* < 0.05, and ns denotes *p* > 0.05. The boxes denote interquartile ranges, and the bottom and top boundaries of the boxes are the 25th and 75th percentiles, respectively. The lower and upper whiskers correspond to the 5th and 95th percentiles, respectively. A horizontal line inside a box denotes the median.

**Figure 2 ijms-26-01740-f002:**
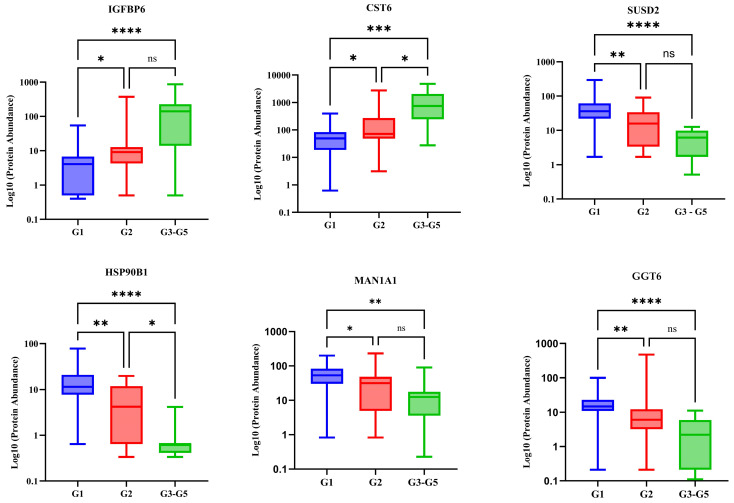
Box and whisker plots of the significantly different urinary protein expression between stages after a Kruskal–Wallis test followed by a Dunn’s test showing the urine proteins expressed in a logarithmic scale (log10) at Stage G1 (eGFR ≥ 90 mL/min/1.73 m^2^), Stage G2 (eGFR 60–89 mL/min/1.73 m^2^), and Stage G3–G5 (eGFR < 60 mL/min/1.73 m^2^). CST6—Cystatin M (CST6), GGT6—glutathione hydrolase 6, SUSD2—sushi domain containing 2, IGFBP6—Insulin-like growth factor binding protein 6, HSP90B1—heat shock protein 90 beta family member 1, MAN1A1—mannosidase alpha class 1A member 1. Asterisks indicate statistically significant differences between the stages, where **** *p* < 0.0001, *** *p* < 0.001, ** *p* < 0.01, and * *p* < 0.05 and ns denotes *p* > 0.05. The boxes denote interquartile ranges, and the bottom and top boundaries of the boxes are the 25th and 75th percentiles, respectively. The lower and upper whiskers correspond to the 5th and 95th percentiles, respectively. A horizontal line inside a box denotes the median.

**Figure 3 ijms-26-01740-f003:**
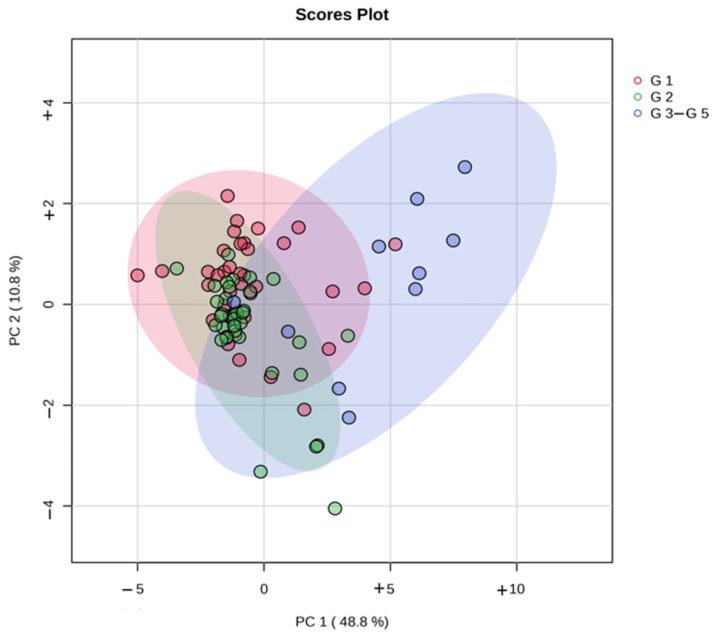
Principal component analysis (PCA) plots obtained for the 38 differential urine proteins across three stages: Stage G1 (eGFR ≥ 90 mL/min/1.73 m^2^), Stage G2 (eGFR 60–89 mL/min/1.73 m^2^), and Stage G3–G5 (eGFR <60 mL/min/1.73 m^2^).

**Figure 4 ijms-26-01740-f004:**
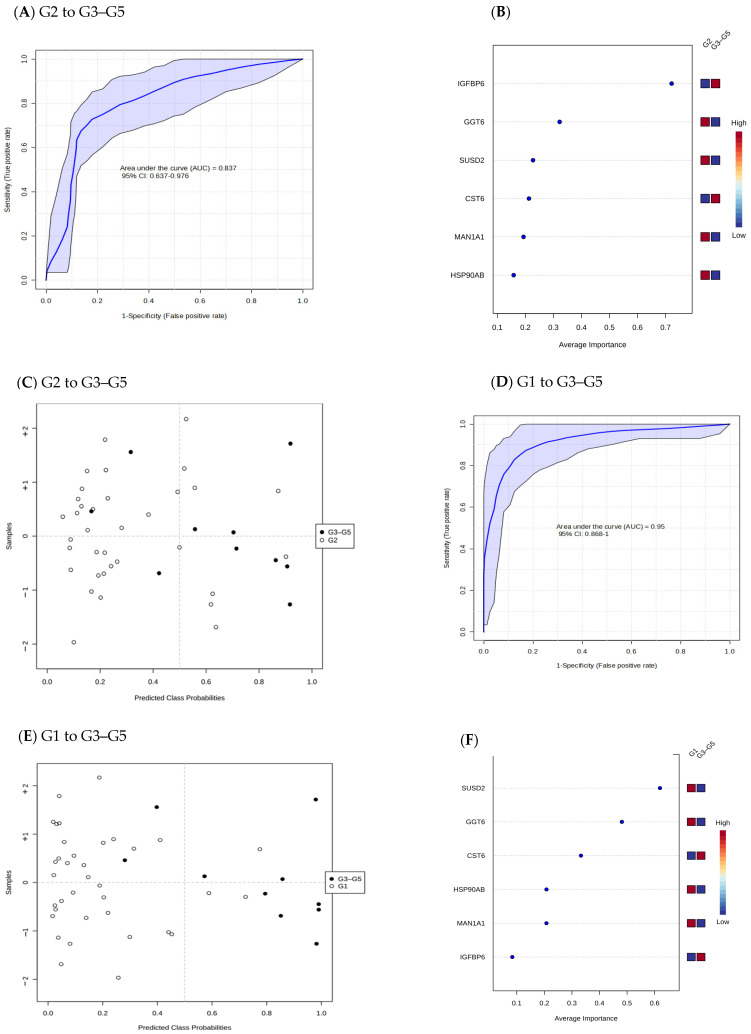
Receiver operating characteristics (ROC) curves for different stages of selected differential abundant proteins (**A**,**D**). The most discriminating urine proteins are shown in descending order of their coefficient scores based on the mean of the importance between groups (**B**,**F**). The coloured boxes indicate whether the protein level is increased (red) or decreased (blue). Cross validation prediction (**C**,**E**) of 6 selected urine proteins. A purple shaded area in an ROC indicates the 95% confidence interval. Stage G1 (eGFR ≥ 90 mL/min/1.73 m^2^), Stage G2 (eGFR 60–89 mL/min/1.73 m^2^), and Stage G3–G5 (eGFR < 60 mL/min/1.73 m^2^).

**Figure 5 ijms-26-01740-f005:**
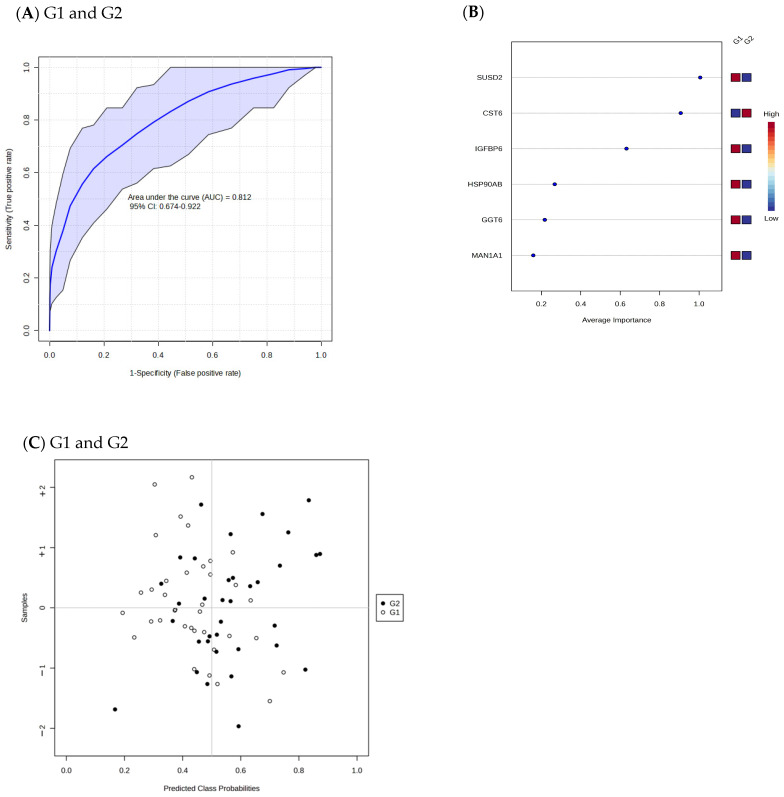
Receiver operating characteristic curves comparing differentially abundant proteins between Stage G1 and Stage G2 (**A**). The most discriminating urine proteins are shown in descending order of their coefficient scores based on the mean of the importance between groups (**B**). The coloured boxes indicate whether the protein level is increased (red) or decreased (blue). Cross validation prediction of 6 selected urine proteins (**C**). A purple shaded area indicates the 95% confidence interval. Stage G1 (eGFR ≥ 90 mL/min/1.73 m^2^), Stage G2 (eGFR 60–89 mL/min/1.73 m^2^), and Stage G3–G5 (eGFR < 60 mL/min/1.73 m^2^).

**Figure 6 ijms-26-01740-f006:**
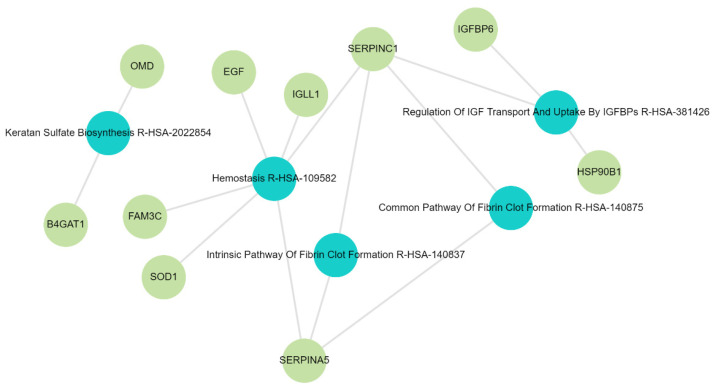
Gene Ontology Reactome 2022 (top 5 pathways). The blue circles indicate Reactome pathways and green circles indicate the genes involved.

**Table 1 ijms-26-01740-t001:** Demographic and clinical characteristics of study participants.

Variable	Total	Stage G1 *N* = 36	Stage G2 *N* = 35	Stage G3–G5 *N* = 10	*p*-Value
Age, years	52 (36–63)	50 (35–62)	50 (35–61)	63 (48–68)	0.167
Sex, female	45/81 (56)	19/36 (53)	19/35 (54)	7/10 (70)	0.671
BMI, kg/m^2^	26 (22–29)	24 (21–27)	27 (23–31)	26 (22–27)	0.195
Serum creatinine (µmol/L)	73 (60–90)	56 (50–69)	79 (72–99)	119 (101–140)	<0.001 *^†#^
eGFR (mL/min/1.73 m^2^)	88 (76–106)	107 (100–117)	84 (75–88)	52 (44–56)	<0.001 *^†#^
uACR (mg/mmol)	0.4 (0.2–2.3)	0.4 (0.2–1.1)	0.3 (0.2–0.8)	6.8 (2.3–50)	<0.001 *^†#^
SBP (mmHg)	131 (121–140)	124 (120–142)	131 (121–138)	137 (128–156)	0.242
DBP (mmHg)	80 (73–86)	79 (62–88)	81 (71–87)	80 (68–89)	0.779
Participants with hypertension	16/72 (22)	4/32 (13)	7/31 (23)	5/9 (56)	0.023 ^†^
Participants with diabetes	19/80 (2.7)	3/36 (8)	11/34 (33)	5/10 (50)	0.004 *^†^
Participants with HIV infection	23/81 (28.4)	12/36 (33.3)	7/35 (20)	4/10 (40)	0.360

BMI: body mass index, SBP: systolic blood pressure, DBP: diastolic blood pressure, eGFR: estimated glomerular filtration rate, uACR: urine albumin:creatinine ratio. Continuous variables are expressed as medians (25th and 75th interquartile range), and categorical variables are expressed as proportions in percent (%). Stage G1 (eGFR ≥ 90 mL/min/1.73 m^2^), Stage G2 (eGFR 60–89 mL/min/1.73 m^2^), and Stage G3–G5 (eGFR < 60 mL/min/1.73 m^2^). * Significant differences between Stage G2 and Stage G3–G5. ^†^ Significant differences between Stage G1 and Stage G3–G5. ^#^ Significant differences between Stage G1 and Stage G2.

**Table 2 ijms-26-01740-t002:** GO biological processes, molecular functions, and pathway analysis of candidate markers.

	Name	Candidate Genes	Adjusted *p*-Value
Biological GO	Retina Homeostasis (GO:0001895)	*IGHG3; B2M; SOD1*	0.0176
	ESCRT III Complex Disassembly (GO:1904903)	*IST1; VTA1*	0.0176
	ESCRT Complex Disassembly (GO:1904896)	*IST1; VTA1*	0.0176
Cellular GO	Intracellular Organelle Lumen (GO:0070013)	*HPX; HSP90AB1; GSN; SERPINC1; OMD; SERPINI1; B2M; HSP90B1; SOD1*	0.0010
	Secretory Granule Lumen (GO:0034774)	*HSP90AB1; GSN; EGF; IST1; FAM3C; B2M*	0.0010
	Collagen-Containing Extracellular Matrix (GO:0062023)	*HPX; SERPINC1; MMRN2; SERPINA5; HSP90B1*	0.0140
	Endocytic Vesicle Lumen (GO:0071682)	*HPX; HSP90B1*	0.0210
Molecular GO	Serine-Type Endopeptidase Inhibitor Activity (GO:0004867)	*SERPINC1; SERPINI1; SERPINA5*	0.0097
	MHC Class II Protein Complex Binding (GO:0023026)	*HSP90AB1; B2M*	0.0291
	Endopeptidase Inhibitor Activity (GO:0004866)	*SERPINC1; SERPINI1; SERPINA5*	0.0291

## Data Availability

The mass spectrometry proteomics data have been deposited in the ProteomeXchange Consortium via the PRIDE [52] partner repository, with the dataset identifier PXD054170.

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
