# Peer review of "Distinct Urinary Proteome Changes Across Estimated Glomerular Filtration Rate Stages in a Cohort of Black South Africans"

_ijms, 2025, doi:10.3390/ijms26041740_

Round 1

Reviewer 1 Report

Comments and Suggestions for Authors

The manuscript addresses the significant topic of protein predictors of CKD. However, in its current form, it is not suitable for publication. The introduction does not provide enough information.  The purpose and significance of the work is not clearly stated. The supplement is supposed to contain supplementary material, and here it appears that more results are in the supplement than in the main text. Before coming across the first figure in the results, the reader is referred back to the supplement 5 times. The authors mention that a weakness of other studies was the small number of participants while they present n=10 for the G3-G5 group. Does it make sense to include such a small group? At the same time, this is a group in which the sex ratio is also disturbed. How does this study provide information on pathophysiological mechanisms?

  • The explanation of abbreviations should come when the abbreviation appears. At the moment they appear at any time throughout the manuscript or do not appear at all.
  • Line 64-65: is there any source to back up this statement
  • Line 68: "1-3- Are you referring to G1-3 or A1-3?
  • Line 73-75: Do you keep talking about the same population as in the previous sentence?
  • Line 78: “Severe” meaning what stage?
  • Line 86: how small?
  • Line 92: Not across all groups
  • Table 1: What is the number 96 at the top of the table? Why are 0.023 and 0.004 bolded but no significance was marked? Three symbols were used for “Significant differences between Stage G1 and Stage G3-5, how do they differ?
  • Is section 2.2. really a result?
  • How many proteins were analyzed?
  • Line 124: which one? 6 or 38?
  • Figure 1: What range has a box and the whiskers? Why log-transformed levels are presented? The axis caption should indicate that it is log-transformed data.
  • Section 2.4 is poorly written.
  • Line 205-207: there is no mention in the text of such proteins
  • Line 211-212: there is no mention in the text of the eGFR correlation
  • Line 285: why were 81 patients chosen in particular? What were the inclusion and exclusion criteria?
  • Line 305: How do four times mean 1600uL?
  • Section 4.5: In what repository is the LCMS data deposited?
  • Section 4.6: Why weren't all groups compared at the same time, but “each with each”?
  • Line 370: The second sentence suggests that they are completely different proteins than in sentence one
  • authors contribution section does not justify the order of authors
  • line 6: according to line 6 ISG and JF contributed equally to this work, according to authors contribution they did not
Comments on the Quality of English Language

some sentences are hard to understand

Author Response

Comment 1: Comments and Suggestions for Authors

The manuscript addresses the significant topic of protein predictors of CKD. However, in its current form, it is not suitable for publication. The introduction does not provide enough information. The purpose and significance of the work is not clearly stated. The supplement is supposed to contain supplementary material, and here it appears that more results are in the supplement than in the main text. Before coming across the first figure in the results, the reader is referred back to the supplement 5 times. The authors mention that a weakness of other studies was the small number of participants while they present n=10 for the G3-G5 group. Does it make sense to include such a small group? At the same time, this is a group in which the sex ratio is also disturbed. How does this study provide information on pathophysiological mechanisms?

Response: Authors would like to thank the reviewer for constructive feedback. We have added more information to the introduction (Line 79-82). In addition, we have moved figures from the supplementary section to the manuscript where appropriate.

 Additionally, we added the following text to better outline the purpose of this work (Line 92-95): ‘Urinary proteomic profiling can help identify biomarkers of kidney dysfunction at the earliest stage of CKD, even before traditional markers such eGFR and albuminuria show noticeable changes. This enables timely intervention and better management of the disease’.

We reanalysed the data to confirm if median age between the groups is different and it was found to be not statistically different and have amended text accordingly (Line 102).

We agree that the advanced group (G3-G5) had small sample size due to the fact this was population based, and participants are unlikely to survive without treatment. We are of view that despite lower numbers, other stages are sufficient power. However, study highlighted included few patients in the early chronic kidney disease (CKD) stages where intervention is likely to yield more results. Our cohort included 36 patients in Stage G1 and 35 patients in Stage 2 compared to five and 9 patients in highlighted studies.  

How does this study provide information on pathophysiological mechanisms?

We have added more information under section gene ontology/pathways explaining the pathways in which differentially abundant proteins are involved such haemostasis which is implicated in the pathophysiological mechanism of CKD (Section 2.4 and line 352-364) .

Comment 2:  The explanation of abbreviations should come when the abbreviation appears. At the moment they appear at any time throughout the manuscript or do not appear at all.

Response:  Thank you for pointing this error. This has been corrected in the abstract and rest of the manuscript.

Comment 3: Line 64-65: is there any source to back up this statement

Response: The sources have been added to back the statement (Line 67-68) and reference list updated accordingly.  

  1. Emrich IE, Pickering JW, Götzinger F, Kramann R, Kunz M, Lauder L, et.al. Comparison of three creatinine-based equations to predict adverse outcome in a cardiovascular high-risk cohort: an investigation using the SPRINT research materials, Clinical Kidney Journal 2024, 17(2). https://doi.org/10.1093/ckj/sfae011

  1. Tofte N, Lindhardt M, Adamova K, Bakker S.J.L, Beige J, Beulens J, et al.. Early detection of diabetic kidney disease by urinary proteomics and subsequent intervention with spironolactone to delay progression (PRIORITY): A prospective observational study and embedded randomised placebo-controlled trial. Lancet Diabetes Endocrinol. 2020, 8. 1016/S2213-8587(20)30026-7

Comment 4: Line 68: "1-3" - Are you referring to G1-3 or A1-3?

Response:  The CKD stages 1-3 refer to CKD G1-3. The text has been modified for clarity. It is now reads ‘In a study comparing CKD stages G1-3 to healthy controls in an Asian…..’. (Line 71)

Comment 5:  Line 73-75: Do you keep talking about the same population as in the previous sentence?

Response: We have combined the two sentences referencing the same study to the repetitiveness (Line 69-74). It now reads ‘In a study comparing CKD stages G1-3 to healthy controls in an Asian population, there were more urinary proteins expressed among healthy controls compared to those with CKD, with lower urinary protein expression with advanced stages of CKD  and urinary proteins such as beta-2 microglobulin, Fetuin A, vitamin-D binding globulin, and alpha-1 microglobulin/bikunin precursor were linked to stages with poorer kidney function’

Commet 6: Line 78: “Severe” meaning what stage?

Response:  The eGFR associated with severe stages was < 29 ml/min/1.73 m2 in this study. This has been included in bracket to indicate the level (Line 84).

Comment 7: Line 86: how small?

Response:  The referenced studies included five and nine patients with stage 1 CKD. It has been included in the text (Line 92-93).

Comment 8: Line 92: Not across all groups

Response: The median age for all groups was not statistically significant using a non-parametric Kruskal Wallis test (Line 102). However, we have modified the statement. It is now read ‘Age and sex were not statistical different between the groups’.

Comment 9: Table 1: What is the number 96 at the top of the table? Why are 0.023 and 0.004 bolded but no significance was marked? Three symbols were used for “Significant differences between Stage G1 and Stage G3-5, how do they differ?

Response:  The number 96 at top of the table corresponds to line 96, this has been adjusted clarity. Thank you for pointing out the error in p values that are bolded but not marked with as significant, this has been corrected (Table 1)

Comment 10: Is section 2.2. really a result?

Response: We are of the view that assessment of performance of quality control and system suitability is key component of this work. We believe that it warrants highlighting it the results section.

Comment 11: How many proteins were analyzed?

Response: A total of 1469 proteins were analysed in this cohort. This following phrase has been included (Line 126):A total of 1469 proteins were identified in this cohort, and 38 urine proteins were significant different across the three stages using the Kruskal-Wallis H test’.

Comment 12: Line 124: which one? 6 or 38?

Response: The phrase refers to six urine proteins that were significantly different between Stage 1 and Stage 2. The sentence has been modified it now reads (Line 132-133): ‘Six urine proteins (CST6, GGT6, SUSD2, IGFBP6, HSP90B1, and MAN1A1) achieved statistical difference between Stage G1 and Stage G2’.

Comment 13: Figure 1: What range has a box and the whiskers? Why log-transformed levels are presented? The axis caption should indicate that it is log-transformed data.

Response: The range of box represented interquartile ranges, and the bottom and top boundaries of boxes are the 25th and 75th percentiles, respectively. Lower and upper whiskers correspond to the 5th and 95th percentiles, respectively. A horizontal line inside the box denotes the median. The y-axis data was expressed in logarithmic (log 10) instead of being log-transformed. The has been corrected and labelled appropriately (Figures 1 and 2).

 One advantage is that a lognormal distribution is easier to see on a logarithmic axis. Another reason to prefer a logarithmic axis is when the values span a large (many orders of magnitude) range of values and otherwise wouldn't really fit on a linear graph.

Comment 14: Section 2.4 is poorly written.

Response: Thank you for your feedback, we have modified this section and hope it is now acceptable. This section has been modified to highlight the genes that contributed to each enrichment pathway. The following phrase has been added (Section 2.4):

 ‘Functional enrichment analysis and gene ontology of DAPs are shown in Figure 6 and Table S4.  To identify functional pathways linked to our proteomic signatures and potentially associated with CKD progression, we conducted pathway enrichment analysis through Enrichr database. The top five significantly enriched terms (adjusted p value < 0.05)  were: haemostasis (SERPINA5, SOD1, IGLL1, FAM3C, EGF, and SERPINC1) as highest mechanism in the pathogenesis of CKD, intrinsic pathway of fibrin clot formation (SERPINA5, and SERPINC1), common pathway of fibrin clot formation (SERPINA5, and SERPINC1), keratin sulfate biosynthesis (B4GAT1, and OMD) and regulation of insulin growth factor 1(IGF-1) transport and uptake by insulin growth (HSP90B1, IGFBP6, and SERPINC1). Furthermore, IGHG3, B2M, and SOD1 were linked to retina homeostasis (GO:0001895) biological process (Table 2). ESCRT III Complex Disassembly (GO:1904903) biological process was associated with IST1 and VTA1 proteins. While molecular function of DAPs was predominantly linked to inhibition of the endopeptidases’.

Table 2 has been added to the manuscript to show genes in enriched pathways.

Comment 15: Line 205-207: there is no mention in the text of such proteins.

Response: These proteins are now mentioned in text and are shown in Figure 1, having been moved from the supplementary data to the manuscript.

Comment 16: Line 211-212: there is no mention in the text of the eGFR correlation.

Response: Thank you for your comment. We have added the following text in Line 207-211:

‘Significant correlations were observed between eGFR and MAN1A1 (r = 0.37, p <0.001), SUSD2 (r = 0.53, p < 0.001), CST6 (r = -0.33, p = 0.003), GGT (r = 0.46, p < 0.001), HSP90B1 (r = 0.43, p < 0.001). While no significant correlation was achieved between IGFBP6 and eGFR (r = 0.05, p = 0.663)’.

Comment 17: Line 285: why were 81 patients chosen in particular? What were the inclusion and exclusion criteria?

Response: We included patients as Stage G3-5 if < 60 mL/min/1.73m² for ≥3 months and excluded those without confirmed low eGFR for Stage G3-5 category. This has been clarified further (Line 386-387)

Comment 18: Line 305: How do four times mean 1600uL?

Response: This has been corrected by the remove ‘four times’ phrase (Line 404). The phrase reads ‘Briefly, 1600 μL of ice-cold 80% acetone was added to 400 μL of urine’.  

Comment 19: Section 4.5: In what repository is the LCMS data deposited?

Response: Data will be deposited into ProteomeXchange Consortium via PRIDE partner with identifier PXD054170 (Line 499-500)

Comment 20: Section 4.6: Why weren't all groups compared at the same time, but “each with each”?

Response:  We performed Kruskal Wallis test to identified differential abundant proteins across all. This was followed by post-hoc (Dunn’s test) analysis. This comparison contrasts Stage G1 (the early stage) with Stage G3-G5 (the later stages). By grouping G3, G4, and G5 together, the analysis aims to look at the differences between the initial stage and the more advanced stages. This could help to identify how the parameter of interest changes as the condition progresses from early to more advanced stages.

Comment 21: Line 370: The second sentence suggests that they are completely different proteins than in sentence one

Response: This sentence has been modified to eliminate ambiguity.

It is now reads (Line 468-469): In conclusion, we identified six urine proteins that were differentially abundant between 60-89 and ≥ 90 eGFR stage. These proteins, combined as part of this study, also provided a 6-protein model that could discriminate between early and late CKD stages with high predictive accuracy.

Comment 22: authors contribution section does not justify the order of authors

Response: This has been corrected to reflect equal contribution.

Comment 23: line 6: according to line 6 ISG and JF contributed equally to this work, according to authors contribution they did not

Response: This has been amended as above comment.

Reviewer 2 Report

Comments and Suggestions for Authors

Dear authors,

1.        The reviewer believes that the section on gene ontology (GO) analysis lacks detailed discussion of the specific results and their significance. Including a more thorough explanation of the GO analysis results would make it easier for readers to understand. For example, it would be helpful to provide prior explanations about which genes are involved in which pathways.

2.        The phrase "there was no absolute separation" is vague. It would be more convincing to provide an explanation of what this lack of complete separation means, such as indicating that there were some errors in classification. Additionally, if the absence of absolute separation suggests challenges in diagnosis, it would be helpful to briefly mention that point.

3.        The authors mention that some results are statistically significant, but addressing their clinical significance would help readers better understand the findings. For example, adding an explanation of how the variation in proteins correlates with kidney function markers would clarify the practical implications of the results.

4.        The authors mention the limitation of having a limited number of patients with low eGFR and conducting the study from a single cohort. However, it would be beneficial to discuss more about the impact of this limitation on the study results. Specifically, discussing how the sample size and cohort selection may influence the generalizability of the findings would provide a deeper understanding of the reliability of the study.

Author Response

Comment 1: The reviewer believes that the section on gene ontology (GO) analysis lacks detailed discussion of the specific results and their significance. Including a more thorough explanation of the GO analysis results would make it easier for readers to understand. For example, it would be helpful to provide prior explanations about which genes are involved in which pathways.

Response: This section has been modified to highlight the genes that contributed to each enrichment pathway (Section 2.4). The following phrase has been added: Functional enrichment analysis and gene ontology of DAPs are shown in Figure 6 and Table S4.  To identify functional pathways linked to our proteomic signatures and potentially associated with CKD progression, we conducted pathway enrichment analysis through Enrichr database. The top five significantly (using adjusted p value < 0.05) enriched terms were: haemostasis (SERPINA5, SOD1, IGLL1, FAM3C, EGF, and SERPINC1) as highest mechanism in the pathogenesis of CKD, intrinsic pathway of fibrin clot formation (SERPINA5, and SERPINC1), common pathway of fibrin clot formation (SERPINA5, and SERPINC1), keratin sulfate biosynthesis (B4GAT1, and OMD) and regulation of insulin growth factor 1(IGF-1) transport and uptake by insulin growth (HSP90B1, IGFBP6, and SERPINC1). Furthermore, IGHG3, B2M, and SOD1 were linked to retina homeostasis (GO:0001895) biological process (Table 2). ESCRT III Complex Disassembly (GO:1904903) biological process was associated with IST1 and VTA1 proteins. While molecular function was predominantly linked to inhibition of endopeptidases.

Table 2 has been added to the text to show gene ontology results.

Comment 2: The phrase "there was no absolute separation" is vague. It would be more convincing to provide an explanation of what this lack of complete separation means, such as indicating that there were some errors in classification. Additionally, if the absence of absolute separation suggests challenges in diagnosis, it would be helpful to briefly mention that point.

Response. This has been noted and this phrase has been modified to (Line 223-224): ‘There was no absolute separation, but most of the samples could be distinguished correctly, especially between Stage G1 and Stage G3 - G5 suggesting misclassification of some samples in this cohort’

Comment 3: The authors mention that some results are statistically significant, but addressing their clinical significance would help readers better understand the findings. For example, adding an explanation of how the variation in proteins correlates with kidney function markers would clarify the practical implications of the results.

Response: Thank you for pointing out these shortcomings. We have added the following phrase to explain the clinical significance of the correlation with traditional markers

Line 299-302: ‘In the advanced stages of CKD, CST6 levels were significantly higher than in the early stages and correlated negatively with eGFR suggesting that kidney function declines (indicated by increased creatinine), CST6 levels increase. This relationship could indicate that CST6 might be involved in kidney injury or dysfunction, potentially serving as a marker for kidney damage or impairment’.

Line 319-321: ‘In this study, MAN1A1 and SUSD2 gradually decreased with decreasing kidney function and levels were lower increasing severity of kidney functioning suggesting that these may play a role in kidney function or repair. Their reduced levels in patients with high uACR could reflect a deteriorating kidney function’.

Line 346-347): ‘Since CKD is a state of oxidative stress, it may therefore be associated with a depletion of GGT6’.

Comment 4: The authors mention the limitation of having a limited number of patients with low eGFR and conducting the study from a single cohort. However, it would be beneficial to discuss more about the impact of this limitation on the study results. Specifically, discussing how the sample size and cohort selection may influence the generalizability of the findings would provide a deeper understanding of the reliability of the study.

Response: In clinical practice, individuals with advanced CKD can easily be identified by traditional markers. The focus of this study was on identification of urine proteins that are associated with early kidney function decline, where early intervention is likely to yield positive outcomes. Therefore, we do not believe that small sample size in patients with advance CKD classification will have effect on the outcomes.

 We have added a sentence explanation this (Line 366-369). Since the study was conducted from a single cohort, we have discussed that the results may not be generalisable.

Round 2

Reviewer 1 Report

Comments and Suggestions for Authors